# Estimation of Soil Organic Carbon Using Vis-NIR Spectral Data and Spectral Feature Bands Selection in Southern Xinjiang, China

**DOI:** 10.3390/s22166124

**Published:** 2022-08-16

**Authors:** Zijin Bai, Modong Xie, Bifeng Hu, Defang Luo, Chang Wan, Jie Peng, Zhou Shi

**Affiliations:** 1College of Agriculture, Tarim University, Alar 843300, China; 2College of Horticulture, Gansu Agricultural University, Lanzhou 730070, China; 3Department of Land Resource Management, School of Tourism and Urban Management, Jiangxi University of Finance and Economics, Nanchang 330013, China; 4College of Mechanical and Electrical Engineering, Tarim University, Alar 843300, China; 5Institute of Applied Remote Sensing and Information Technology, College of Environmental and Resource Sciences, Zhejiang University, Hangzhou 310058, China

**Keywords:** Vis-NIR spectroscopy, SOC, deep learning, spectral feature bands selection

## Abstract

Soil organic carbon (SOC) plays an important role in the global carbon cycle and soil fertility supply. Rapid and accurate estimation of SOC content could provide critical information for crop production, soil management and soil carbon pool regulation. Many researchers have confirmed the feasibility and great potential of visible and near-infrared (Vis-NIR) spectroscopy in evaluating SOC content rapidly and accurately. Here, to evaluate the feasibility of different spectral bands variable selection methods for SOC prediction, we collected a total of 330 surface soil samples from the cotton field in the Alar Reclamation area in the southern part of Xinjiang, which is located in the arid region of northwest China. Then, we estimated the SOC content using laboratory Vis-NIR spectral. The Particle Swarm optimization (PSO), Competitive adaptive reweighted sampling (CARS) and Ant colony optimization (ACO) were adopted to select SOC feature bands. The partial least squares regression (PLSR), random forest (RF) and convolutional neural network (CNN) inversion models were constructed by using full-bands (400–2400 nm) spectra (R) and feature bands, respectively. And we also analyzed the effects of spectral feature band selection methods and modeling methods on the prediction accuracy of SOC. The results indicated that: (1) There are significant differences in the feature bands selected using different methods. The feature bands selected methods substantially reduced the spectral variable dimensionality and model complexity. The models built by the feature bands selected by CARS, PSO and ACO methods showed the different potential of improvement in model accuracy compared with the full-band models. (2) The CNN model had the best performance for predicting SOC. The R^2^ of the optimal CNN model is 0.90 in the validation, which was improved by 0.05 and 0.04 in comparison with the PLSR and RF model, respectively. (3) The highest prediction accuracy was archived by the CNN model using the feature bands selected by CARS (validation set R^2^ = 0.90, RMSE = 0.97 g kg^−1^, RPD = 3.18, RPIQ = 3.11). This study indicated that using the CARS method to select spectral feature bands, combined with the CNN modeling method can well predict SOC content with higher accuracy.

## 1. Introduction

Soil organic carbon (SOC) pool is the most essential component of the global carbon pool; accurate information of SOC has important implications for the global carbon cycle and climate change. In addition, SOC is a vital indicator of soil quality and is related to many fundamental soil ecosystem functions, such as pollutant uptake and retention, and carbon dioxide storage [1]. Accurately, rapidly and economically monitoring SOC is beneficial for a timely understanding of the dynamic changes of SOC. The conventional methods of SOC determination are not only time-consuming, laborious and environmentally polluting, but also use some chemical reagents, which are harmful to human health in the process of chemical analysis, which make it far from ideal for the demand of rapid monitoring of SOC. In recent years, Vis-NIR spectral techniques based on spectroscopic principles have been proven reliable and accurate in estimating SOC content as fast, non-destructive and economical alternative for traditional method [2]. In addition, the Vis-NIR spectral techniques can meet the current demand for quantitative monitoring of SOC with no chemical reagents and no environment pollution [3].

Previously, most studies have used soil reflectance in the Vis-NIR full-band [4,5,6]. The reflectance of soil samples in Vis-NIR region contains information from iron oxides, soil color, and water as well [7]. Different soil properties have their own specific band spectral characteristics. Thus, using full-band soil reflectance to estimate soil constituent content results in redundant information [8,9]. In order to improve the efficiency and accuracy of prediction of soil properties using Vis-NIR spectral, scholars introduced various feature band selection algorithms to remove noise and extract feature bands. Among which, Bao et al. [10] used the competitive adaptive reweighted sampling (CARS) method combined with first-order differentiation of reflectance to develop a model for organic matter prediction with best results. Liu et al. [11] transformed the spectral data by continuous removal, and then the RF model of soil organic matter built by the CARS method was used to select the characteristic bands, which obtained a R^2^ of 0.96 for their prediction. Sun et al. [12] used hyperspectral satellite data and a PLSR model whose bands selected by Genetic algorithm (GA), which can better estimate the SOM content.

Mathematical modeling techniques have been widely used in the field of soil spectroscopy, and among which, partial least squares regression (PLSR) is one of the most widely used methods [13]. The PLSR is a typical linear modeling approach that has been broadly used for SOC estimation [8]. In addition, machine learning algorithms extend their performance to upgrade the accuracy of soil attribute estimation [14]. However, the performance of machine learning models largely depends on the quality of the spectral data, which is influenced by preprocessing methods and the numbers of samples [15]. In recent years, deep learning methods, especially the Convolutional Neural Networks (CNN), have been introduced in spectral data modeling. The CNN is a computational method that extracts features through the convolution and pooling layers. It is suitable for spectral data modeling because of their characteristics of processing high-dimensional data, extracting features, parameter sharing and multiple kernels. [16]. For example, Veres et al. [17] used spectral data combined with 1D-CNN for the first time to predict some soil properties. Since then, some researchers have also used deep learning models based on soil spectral libraries to predict soil properties. Transfer learning was also used to predict the mineral soil properties [18]. Some studies have also converted raw spectra to 2D spectra to predicted some soil properties [19]. Although deep learning methods have been reported extensively in predicting soil properties, the feasibility of deep learning methods for SOC spectral prediction is still largely remains unclear.

To bridge this gap, therefore, this study mainly aims to: (1) screening an optimal SOC feature band selection method; (2) optimize a preferably modeling method of SOC prediction; (3) explore the potential of hyperspectral techniques for quantitative detection of SOC, especially the soil with low organic carbon content in arid and semi-arid regions.

## 2. Materials and Methods

### 2.1. Study Area

The Alar Reclamation (80°30′–81°58′ E, 40°22′–40°57′ N) is a typical artificial oasis located in the southern part of Xinjiang, the arid region of northwest China (Figure 1). It has an area of 6256.68 km^2^. The terrain in the study area inclines from northwest to southeast. The study area has an annual precipitation of 40.1 mm to 82.5 mm and an average annual evaporation of about 1876–2558 mm, with high summer temperatures and plenty of light and heat throughout the year. Additionally, it has high temperatures during the day, low temperatures at night, and mostly dusty weather. The main soil types are brown desert soil, salt soil, and sandy loam, which are poor soils. Cotton, wheat, red dates and other crops are mainly planted, especially the cotton covers more than 80% of the area, so the cotton is the most important cash crop in the study area [20].

### 2.2. Soil Sampling and Laboratory Analysis

A total of 330 surface soil (0–20 cm) samples were collected in the study area using a random sampling method from October to November 2019. The weight of the sample is approximately 1 kg. The soil samples were then placed in sealed bags and taken back to the laboratory to dry naturally. After standardization in the laboratory and sieving through 2 mm and 0.15 mm, they were used for Vis-NIR spectroscopy and SOC determination, respectively.

### 2.3. Soil Spectra Collection and Pre-Processing

Soil samples were measured by Vis-NIR spectroscopy using ASD FieldSpec geophysical spectrometer (Fieldspec4, Analytical Spectral Devices, Inc., Boulder, CO, USA). The instrument parameters are shown in the literature [21]. The soil sample is placed in a black box and measured with the spectrometer’s built-in light contact probe. The spectra were measured 10 times for each soil sample and the average of the spectral reflectance was taken as the spectral value of the soil sample. Vis-NIR band to remove the noise band (350–399 nm and 2401–2500 nm), and retained the 400–2400 nm band, which is then Savitzky-Golay Smoothed.

### 2.4. Sample Selection Algorithm

The SPXY, developed from the Kennard-Stone (KS) method, combines independent and response variables to select a representative sample as opposed to using K-S alone. It combines the independent (*x*) and dependent variables (*y*), and for each pair (m,n) of samples, the target composition distance $D_{y}\left(m,n\right)$ can be calculated [22]. In order to make the distribution of samples in the x and the y equally important, distances $D_{x}\left(m,n\right)$ and $D_{y}\left(m,n\right)$ are divided by their maximum values in the data set. By this way, a normalized xy distance is calculated as follows:(1)$$D_{xy}\left(m,n\right)=\frac{D_{x}\left(m,n\right)}{max_{m,n\in \left[1,N\right]\;}D_{x}\left(m,n\right)}+\frac{D_{y}\left(m,n\right)}{max_{m,n\in \left[1,N\right]\;}D_{y}\left(m,n\right)}$$
where *D_x_(m,n)* and *D_y_(m,n)* is Euclidean distance of samples *m* and *n*. $max_{m,n\in \left[1,N\right]\;}D_{x}\left(m,n\right)$ and $max_{m,n\in \left[1,N\right]\;}D_{y}\left(m,n\right)$ means the maximal distance in *x* and the *y*, respectively. The symbol *N* indicates the number of all samples.

### 2.5. Spectral Feature Bands Selection Algorithm

The feature bands selection method could improve the predictive power and modelling efficiency of the model, which enhances the stability of the model and can provide a better explanation of the process of generating data [23]. It takes into account the interactions between spectral variables to remove the effects of uncorrelated or noisy variables [24,25,26]. In this study, three spectral feature band selection methods (CARS, PSO, ACO) were applied to select SOC feature bands from the full-band spectra.

The CARS algorithm uses an exponentially decreasing function to remove the wavelengths with small regression coefficients, and then builds a PLS model based on the new subset, and selects the smallest root mean square error of cross validation (RMSECV) subset as the characteristic wavelengths after multiple calculations [27].

The Particle Swarm optimization (PSO) is an optimization algorithm that simulates a bird’s random search for food. The algorithm searches by following the optimal particles in space to achieve parameter search, which has the advantage of simplicity and convenience compared to the grid search method [28].

The PSO evaluates the particles by position, velocity and fitness, and the fitness function is the objective to be optimized. The formula of the position and velocity about particles are as follows:(2)$$X_{id}\left(n+1\right)=X_{id}\left(n\right)+V_{id}\left(n+1\right)$$
(3)$$V_{id}\left(n+1\right)=wV_{id}\left(n\right)+c_{1}r_{1}\left(P_{id}\left(n\right)-X_{id}\left(n\right)\right)+c_{2}r_{2}\left(P_{gd}\left(n\right)-X_{id}\left(n\right)\right)$$
where X is the particles position, V is the particles velocity, P is the particle optimal position, *P_g_* is the best position of the particle population, $c_{1}$ and $c_{2}$ are learning factors, $r_{1}$ and $r_{2}$ are random numbers with values in [0,1], i indicates the particle, d indicates the dimension, n indicates the number of cycles, w represents the inertia weight, reflecting the ability to take on the previous particle velocity.

The Ant colony optimization (ACO) simulates ants releasing a secretion, called pheromone, on the paths they walk as they search for food sources [29]. The ants behind determine the direction of foraging based on the pheromones left in the path. When a path is shorter, ants walking on that path will leave more pheromones behind, which will attract more ants to choose that path. But when the number of ants walking through the path is higher, the number of pheromones left behind is also higher, creating a positive feedback mechanism. The ant colony algorithm wavelength selection is to optimize the accuracy of the soil spectral prediction model as the goal, simulate the foraging behavior of ants, and seek the characteristic wavelength according to the change of pheromones on the path.

### 2.6. Modeling Development

#### 2.6.1. PLSR

The PLSR is an analytical method that combines principal component regression, multiple linear regression and typical correlation analysis. The main study of PLSR is to develop a linear model of the independent variables in the large number of two sets of highly linearly correlated variables. It could reduce the dimensionality of the independent variables and eliminate autocorrelation between spectral bands. Therefore, the PLSR is the most classical linear regression algorithm in soil Vis-NIR spectroscopy [30,31].

The number of latent variables is the most critical parameter in the PLSR algorithm. In this study, the optimal number of latent variables was determined by the leave-one-out cross-validation method (LOOCV) [32]. The maximum value of R^2^ corresponds to the optimal number of latent variables.

#### 2.6.2. RF

The random forest (RF) refers to a classifier that uses multiple decision trees to train and predict samples. The RF requires low hyperparameter settings [33]. It randomly selects the nodes of the decision tree to divide the features, which allows efficient training of the model when the sample features are of high dimension, and is suitable for any dataset [34]. The RF implementation is relatively simple and it is insensitive to the absence of some features.

In the RF model, the number of decision trees (ntree) and the number of branches per tree (mtry) are key parameters in determining the accuracy of the model [35]. The mtry refers to the number of input variables, which is generally smaller than the total number of variable combinations. The value of ntrees is used to regulate the computational volume and running time of the model. In this study, we set the value of ntrees to 500. The value of mtry was taken from 1 to the number of independent variables [36]. The best parameters were selected by cross-validation the root mean square error minimum (RMSE_CV_), and the RF model with the minimum RMSE_CV_ was the best model.

#### 2.6.3. CNN

The CNN is one of the first proposed deep learning algorithms; it is widely used for its powerful feature characterization capability [17]. Convolution is applied to spectral data, using convolution filters to extract spectral features. Each unit in the convolution layer is connected to a local feature in the spectral feature. The final size of the spectral features is determined by the kernel size, the stride and the padding. The pooling layer is used to reduce the dimensionality of the spectral features to improve the computational efficiency and to preserve important information while preventing overfitting of the network [37].

The AlexNet model has successfully applied the ReLU activation function and Dropout layer in CNN, and the research and application of deep learning has gained wide attention. The ReLU activation function can perform nonlinear mapping of neurons, and the Dropout layer can effectively prevent the overfitting phenomenon, so that CNN can complete classification and regression more accurately. The ReLU is computationally inexpensive and it converges faster than the sigmoid and tanh functions [38].

The CNN in this paper consists of seven trainable layers, as illustrated in Figure 2. The convolutional layers contain 16 filters with a size of 8 and a step size of 1. After that, the number of filters is increased by a factor of 2 for each convolutional layer, while the other parameters are kept constant. The pool size is set to 2 for the first layer and 4 for the later ones. The final output feature map of the pooling layer is squashed and connected to the fully connected layer as input. Some layers in the network are shared to allow for multitasking settings. The weights are updated using the Adam optimizer. The training results of CNN are influenced by many hyperparameters. Here, we use Bayesian optimization approach to fine-tune hyperparameters such as dropout, learning rate, and stopping criterion [39].

### 2.7. Indicators for Model Evaluation

The R^2^, root mean square error (RMSE), relative analysis error (RPD) and the ratio of performance to the interquartile range (RPIQ) were used for assessed the model performances. For the RPD, RPD > 2 indicates that the model is better, when 1.4 < RPD < 2, the model is average, and when RPD < 1.4, the model is not credible [40]. For RPIQ, when RPIQ ≥ 2.2, the model has good predictive ability; 1.7 ≤ RPIQ < 2.2, the model has average predictive ability; RPIQ < 1.7, indicating low model reliability [41]. Referring to the R^2^ value when both models have the same RPD value, the model value better explains the fit of the data with the higher R^2^. When both models have the same RPD and R^2^ values, the model with the lower RMSE is better at predicting and validating the data with reference to the RMSE value.
(4)$$R^{2}=\frac{\sum_{i=1}^{n}{\left({\hat{y}}_{i}-{\bar{y}}_{i}\right)}^{2}}{\sum_{i=1}^{n}{\left(y_{i}-{\bar{y}}_{i}\right)}^{2}}$$
(5)$$\mathrm{RMSE}=\sqrt{\frac{\sum_{i=1}^{n}{\left({\hat{y}}_{i}-y_{i}\right)}^{2}}{n}}$$
(6)$$\mathrm{RPD}=\frac{SD_{y_{i}}}{\mathrm{RMSE}}$$
(7)$$\mathrm{RPIQ}=\frac{Q3-Q1}{\mathrm{RMSE}}$$
where $\hat{y}$ is the predicted value; $\bar{y}$ is the average of the observed values; $y_{i}$ is the observed values; and n is the number of samples. Q3−Q1 is the difference between the third and first quartiles of observations.

## 3. Results

### 3.1. Descriptive Statistics of SOC

The soil samples were divided into calibration and validation according to the SPXY sampling method with the ratio of 2:1 (Table 1). For the entire data set, SOC content ranged from 0.98 to 20.49 g kg^−1^ with a mean value of 6.54 g kg^−1^ and a coefficient of variation of 47.14%. This indicates that the SOC content in the study area has a significant spatial variability and a larger soil variability can improve the predictive power of the model [42]. Similarly, as revealed by the skewness and kurtosis values of the SOC descriptive statistics Table 1, the SOC content data shows a non-normal distribution.

### 3.2. Feature Band Selection Based on CARS, PSO and ACO

As shown in Figure 3, the number of feature bands screened by the three algorithms are different. The number of feature bands, selected by CARS is the least, followed by ACO and PSO. The feature bands of CARS selecting shows segmented distribution in the full-band and less distribution in the visible region. The optimal subset of variables obtained is mainly concentrated in the range 1400–2400 nm, and the soil spectral features in this range are mainly caused by C=O, CH, Al-OH and OH fundamental frequency vibrations and their combined frequencies and octave vibrations [43,44], but the feature bands selected with ACO and PSO are evenly distributed in the full-band. The full-band is selected by CARS, and a total of 90 feature bands are selected, accounting for 4.5% of the full-band number. 156 and 185 feature bands were screened out by ACO and PSO, accounting for 7.8% and 9.2% of the total number of bands, respectively. The number of bands removed after processing by the three algorithms reached more than 90% (Figure 3), which could significantly reduce the input volume of the model, autocorrelation between bands, and improve the model calculation efficiency.

### 3.3. Correlation Analysis between SOC and Spectrum

The correlation curve between the spectral reflectance and SOC content over the wavelength range Vis-NIR is presented in Figure 4. It indicated that in the range of 400–715 nm, SOC and spectra showed a positive correlation with a maximum correlation coefficient of 0.172. After 715 nm, there was a negative correlation between SOC and Vis-NIR spectral reflectance. Among them, the correlation between SOC and spectrum reached a highly significant level in the range of 1235–2400 nm. The highest correlation between SOC and Vis-NIR spectra was reached at 2060 nm (r = −0.337). Additionally, the correlation coefficients of the characteristic bands selected by CARS, PSO and ACO with organic carbon (Figure 4). Specifically, the number of bands that reached highly significant levels for the characteristic bands selected by the three methods differed. The characteristic bands selected by CARS reached a highly significant level accounting for 67% of the total number of bands selected, indicating that the bands selected by CARS were beneficial for model building. But the number of bands that reached highly significant levels with PSO and ACO selected feature bands accounted for 52% and 64% of the total number of feature bands, respectively. In terms of the characteristic band correlation level, CARS was more effective than the other two methods, followed by ACO and PSO. However, it was not the bands with high correlation levels that were most suitable for model construction. As the relationship between spectra and SOC was a nonlinear one, which might result in some of the bands screened not reaching a highly significant level.

### 3.4. SOC Estimation Models Performance

The R^2^ in the validation sets of the models built by the CARS, PSO, and ACO methods selected in the characteristic bands are all above 0.82, compared to the models built by the full-band spectra (validation set R^2^ ≤ 0.81) (Table 2). The prediction models based on the selected feature bands all show robust performance, with overall results better than the full-band spectral models. Otherwise, the prediction accuracy varied widely between the models built by the bands selected by different feature band selection methods. The R^2^ of the model make prediction based on the feature bands screened by CARS, PSO, and ACO in calibration set ranged from 0.90 to 0.94, 0.85 to 0.87, and 0.86 to 0.88, respectively. The R^2^ of the validation set ranged from 0.85 to 0.90, 0.82 to 0.85, and 0.83 to 0.86, respectively. It reveals that choosing the appropriate feature band selection method was crucial for modeling. Thus, we could conclude that the data sets screened by the three methods for modeling were CARS > ACO > PSO in order of effectiveness.

The performance of the models constructed by different algorithms also showed clear differences. The R^2^ ranges for the PLSR, RF and CNN models in calibration data sets were 0.84–0.90, 0.83–0.93, 0.84–0.94, and the RMSE ranges are 0.98–1.21 g kg^−1^, 0.87–1.26 g kg^−1^, 0.72–1.20 g kg^−1^, respectively. The R^2^ ranges of the PLSR, RF and CNN models in validation set were 0.80–0.85, 0.80–0.86, 0.81–0.90, and the RMSE ranges were 1.21–1.33 g kg^−1^, 1.17–1.32 g kg^−1^, 0.97–1.31 g kg^−1^, correspondingly. Meanwhile, RPD ranges of the PLSR, RF and CNN models were 2.23–2.54, 2.25–2.62, 2.27–3.18, and the RPIQ ranges of the PLSR, RF and CNN models were 1.90–2.49, 1.96–2.57, and 2.02–3.11, respectively. The CNN model had the best performance followed by RF, and the PLSR method had the worst performance. According to the evaluation criteria of RPD and RPIQ, the PLSR, RF, and CNN models all had high prediction accuracy with the characteristic bands selected by CARS. The model built with the feature bands selected by CARS combined with CNN has prediction values closer to the 1:1 line. This indicates that the model has a high prediction accuracy, which is the optimized SOC estimation model (Figure 5).

## 4. Discussion

### 4.1. Effects of Feature Band Selection Algorithm on Model Performance

The results obtained from the study show that selecting effective spectral feature band is a key basic feature for building a robust model. The full-band data contains a lot of redundant information, and variable filtering can remove some redundant bands [45]. The number of feature band selected by the three algorithms are different. The CARS, PSO, and ACO were introduced to select the feature band of SOC, and the models based on all three feature band selection algorithms had better prediction performance compared with the models built based on full-band. A robust and highly accurate model can be obtained with these selected feature variables. However, our results showed that PSO and ACO was not as effective as CARS. This is mainly because the PSO algorithm is susceptible to the greater influence of inertia weights w, learning factors *c*_1_ and *c*_2_. The inertia weights w are small in favor of exploiting the local search capability of the particle swarm algorithm. A large inertia weight will be biased to exploit the global search capability of the PSO. The learning factors *c*_1_ and *c*_2_ also serve to balance the ability of local search and global search. The larger the learning factors *c*_1_ and *c*_2_, the stronger the local search capability and the more favorable the convergence of the algorithm [46].

The ACO algorithm has positive feedback, which makes it give a better convergence rate. However, if the algorithm starts with a suboptimal solution, then positive feedback can cause the suboptimal solution to dominate quickly, causing the algorithm to fall into a local optimum. In addition, there are many parameters in the ACO algorithm, which may correlate with each other. The selection of parameters relies on experience and experimentation, and unsuitable initial parameters can reduce the algorithm’s ability to find the best [47]. Therefore, how to reasonably set the values of the parameters is a key issue to be solved in the subsequent study.

The CARS model uses adaptive reweighted sampling (ARS) to select the wavelengths with the highest regression coefficients in the PLS model. It removes the wavelengths with the lowest weights, and uses cross-validation to select the subset of wavelengths with the lowest RMSE_CV_ [8]. When selecting variables, CARS focuses on considering the interactions between variables and can effectively eliminate uninformative variables. The predictive performance of the model is improved compared to the use of full-band [11]. It is an effective feature band selection method for eliminating band redundancy without the restriction of interval selection [10]. The CARS model solves the problems of large wavelength, unstable model accuracy, and large computational effort in SOC prediction [48]. The prediction accuracy of the model built from 90 feature band selected from 2001 bands (R^2^ > 0.85) is significantly higher than that of the model built based on full-band (R^2^ < 0.81), verifying the feasibility of CARS in Vis-NIR prediction of SOC.

### 4.2. Performance Comparison of CNN and Traditional Prediction Models

To evaluate the predictive performance of CNN, we compared it with classical linear regression PLSR and machine learning RF (Table 2). Overall, the CNN model showed best performance and the PLSR and RF models showed moderate performance. Considering the advantage of avoiding covariance, the PLSR model was successfully used to develop estimates of soil properties by many studies [49,50,51]. However, the performance of the PLSR model is easily negatively affected by others factors such as soil texture and color [52]. In this study, the PLSR model built with the SOC feature band selected by CARS had a high accuracy (R^2^ = 0.85). Therefore, the spectral data can be effectively preprocessed to improve the model accuracy in the subsequent study. 

The RF has the advantages of simple modeling process, fast computational speed, and small computational cost [11,53,54]. The performance of the RF model in this study was not the best, which may be due to the fact that RF can be over- or under-fitted when there is noise in the dataset [53]. When a RF model is constructed, the spectral data can be effectively denoised as a way to improve the modeling accuracy. Secondly, the RF model is similar to a black box; there is no control over the inner workings of the model and the model accuracy can only be controlled by adjusting the parameters. 

In addition, the CNN is currently the fastest growing deep learning algorithm. CNNs, similarly to other deep learning algorithms, are composed of several layers. There are three main hierarchies between the input and output layers: the convolutional layer, the pooling layer and the fully connected layer [17]. CNN was originally designed to handle multi-dimensional data, which makes it well suited to multi-band hyperspectral data. [55]. In soil spectroscopy, Veres et al. [17] used the LUCAS dataset to construct a 2D-CNN model for some soil properties. Subsequently, Padarian et al. [19] converted the spectral data into a 2D spectral map and built a 2D-CNN model for multi-task prediction of soil properties. Generally, the CNN framework is more effective than other neural networks for automatic detection of important features. Therefore, CNN is considered the most powerful and dominant deep learning for remote sensing applications [56]. However, its effectiveness is constrained by factors such as hardware quality, the amount of training data and parameter optimization [57].

In this study, the CNN model clearly outperformed the RF and PLSR models for predicting SOC. The advantage of CNN model is that it does not need to reduce the amount of input or extract relevant information and process the input data separately. However, when modeling with R as the input quantity, CNN is less effective. This may be attributed to the fact that the CNN model needs to optimize more parameters, among which the number of convolutional layers, the objective function and the data distribution are the key parameters affecting the generalization ability of CNN. Therefore, setting the parameters of the CNN correctly is a key step to improve the CNN model.

## 5. Conclusions

Both the feature band selection algorithm and the modeling method have great influence on the prediction accuracy of SOC. The SOC feature bands selected by the CARS, PSO and ACO algorithms were distributed in the whole region of the full-band (400–2400 nm). The number of bands selected were all less than 10% of the total number of bands. The accuracy of the estimation models built using the characteristic bands was improved compared to the full-band models. Among them, the CARS algorithm was the optimal SOC feature band extraction algorithm. In addition, we successfully applied PLSR, RF and CNN models to predict SOC from full-band spectral and different feature band data. The overall accuracy of the CNN model was best. Compared with the RF and PLSR models, the CNN model could significantly reduce the error and the model estimation SOC error rate was lower. The high accuracy and low error of CNN made it a superior model for SOC prediction. Collectively, the CNN model based on the CARS algorithm was the optimal SOC estimation model (validation set R^2^ = 0.90; RMSE = 0.97 g kg^−1^; RPD = 3.18; RPIQ = 3.11).

## Figures and Tables

**Figure 1 sensors-22-06124-f001:**
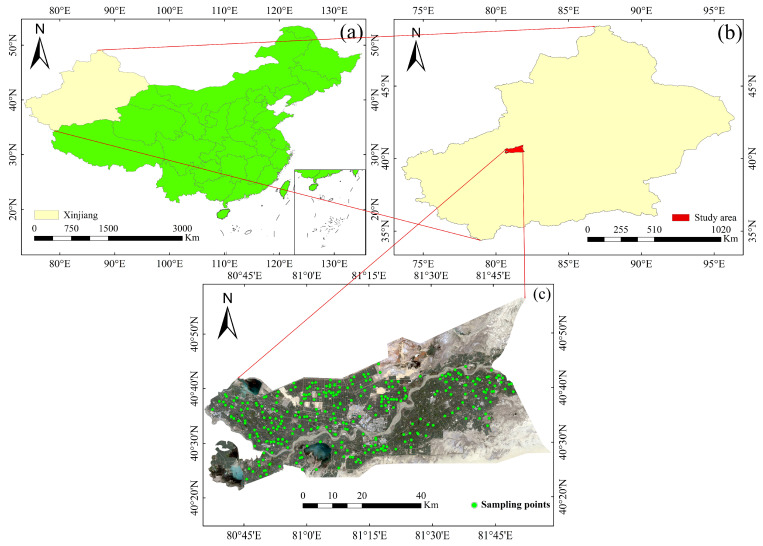
Study area. The location of Xinjiang, China (**a**), The location of the study area (**b**), The distribution of sampling points (**c**).

**Figure 2 sensors-22-06124-f002:**
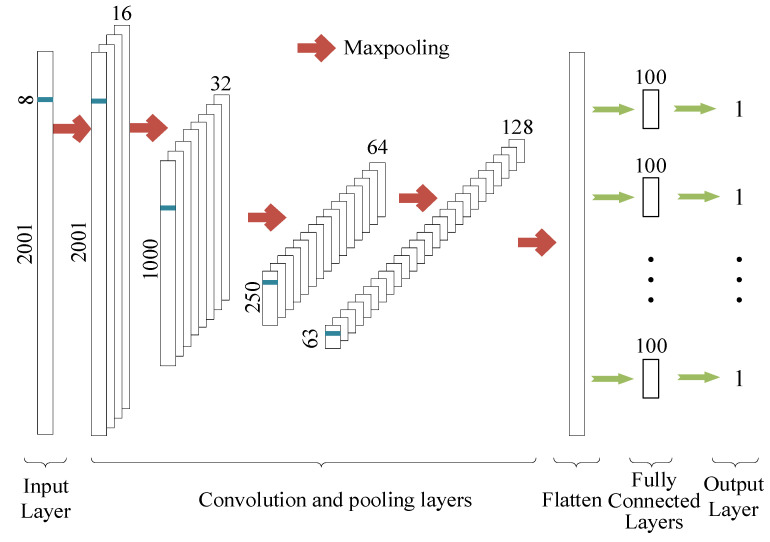
Convolutional Neural Network (CNN) architecture.

**Figure 3 sensors-22-06124-f003:**
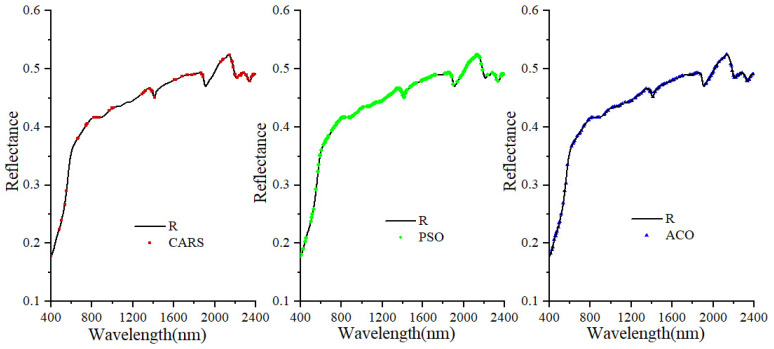
Filtered feature bands based on different algorithms.

**Figure 4 sensors-22-06124-f004:**
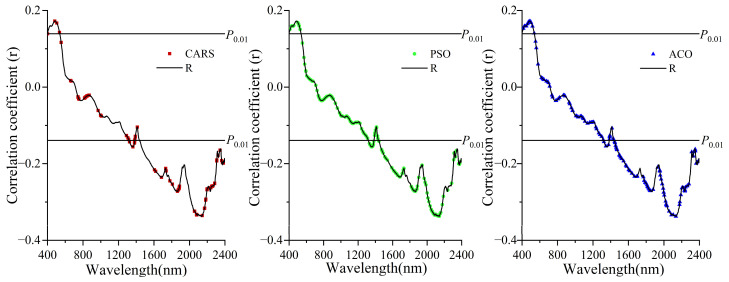
Correlation between soil reflectance and SOC.

**Figure 5 sensors-22-06124-f005:**
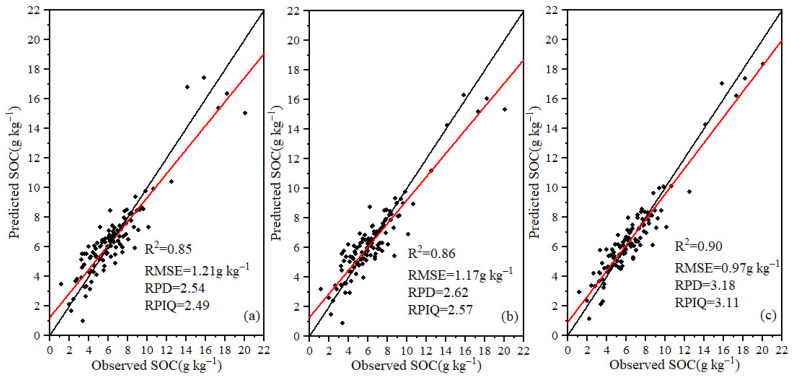
Scatter plots of SOC content estimates using PLSR (**a**), RF (**b**), and CNN (**c**) models built with feature bands selected by CARS algorithm.

**Table 1 sensors-22-06124-t001:** Descriptive statistics of SOC (g kg^−1^).

Sample Set	Number	Range	Mean	SD	CV (%)	Kurtosis	Skewness
Calibration	220	0.98~20.49	6.54	3.08	47.12	5.54	1.92
Validation	110	1.19~20.04	6.54	3.09	47.19	5.53	1.91
Entire	330	0.98~20.49	6.54	3.08	47.14	5.43	1.91

**Table 2 sensors-22-06124-t002:** Prediction accuracy of different models for SOC.

Dataset	Model	Calibration		Validation			
		R^2^	RMSE (g kg^−1^)	R^2^	RMSE (g kg^−1^)	RPD	RPIQ
	PLSR	0.84	1.21	0.80	1.33	2.23	1.90
Full-band	RF	0.83	1.26	0.80	1.32	2.25	1.96
	CNN	0.84	1.20	0.81	1.31	2.27	2.02
	PLSR	0.90	0.98	0.85	1.21	2.54	2.49
CARS	RF	0.93	0.87	0.86	1.17	2.62	2.57
	CNN	0.94	0.72	0.90	0.97	3.18	3.11
	PLSR	0.85	1.20	0.82	1.29	2.32	2.15
PSO	RF	0.87	1.15	0.84	1.24	2.45	2.26
	CNN	0.87	1.16	0.85	1.22	2.53	2.36
	PLSR	0.86	1.14	0.83	1.27	2.39	2.19
ACO	RF	0.87	1.15	0.84	1.26	2.42	2.28
	CNN	0.88	1.09	0.86	1.17	2.57	2.53

## Data Availability

Not applicable.

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
