# Peer review of "Estimation of Soil Organic Carbon Using Vis-NIR Spectral Data and Spectral Feature Bands Selection in Southern Xinjiang, China"

_sensors, 2022, doi:10.3390/s22166124_

Round 1

Reviewer 1 Report

Please refer to the document of review comments for future revision.

Author Response

Dear Editor and Reviewers.
Thank you for your letter and the reviewers' comments on our manuscript entitled "Estimation of soil organic carbon using Vis-NIR spectral data and spectral feature bands selection in Southern Xinjiang, China". The comments made by the reviewers on our manuscript entitled These comments were valuable and helpful in revising and improving our paper, as well as providing important guidance for our research. We have carefully studied these comments and made revisions, which we hope will be accepted by all of you. Please see the attached document for the revised section.

Reviewer 2 Report

Line 28-30. The sentence was repeated twice and need to be more specific.

Line 90. notify the logic: choose a selection method. The section of study purposes need rewritten (Line 90-94).

Figure 1 can be synthesized into an eagle-eye image.

In my knowledge, the formula 6 is wrong.

Line 330 the “a which “should be “which”.

Author Response

(The authors gave the same response as above.)
